# Interstitial Lung Disease Worsens Short- and Long-Term Outcomes of Systemic Rheumatic Disease Patients Admitted to the ICU: A Multicenter Study

**DOI:** 10.3390/jcm10051037

**Published:** 2021-03-03

**Authors:** Lorrain Banuls, Juliette Vanoverschelde, Fanny Garnier, Matthieu Amalric, Samir Jaber, Jonathan Charbit, Kevin Chalard, Marc Mourad, Nacim Benchabane, Racim Benomar, Noemie Besnard, Delphine Daubin, Vincent Brunot, Kada Klouche, Romaric Larcher

**Affiliations:** 1Intensive Care Medicine Department, Lapeyronie Hospital, Montpellier University Hospital, 191, avenue du Doyen Gaston Giraud, 34090 Montpellier, France; lorrain.banuls@gmail.com (L.B.); f-garnier@chu-montpellier.fr (F.G.); amalric.matthieu@gmail.com (M.A.); n-benchabane@chu-montpellier.fr (N.B.); r-benomar@chu-montpellier.fr (R.B.); n-besnard@chu-montpellier.fr (N.B.); d-daubin@chu-montpellier.fr (D.D.); v-brunot@chu-montpellier.fr (V.B.); k-klouche@chu-montpellier.fr (K.K.); 2Radiology Department, Arnaud de Villeneuve Hospital, Montpellier University Hospital, 34090 Montpellier, France; j-vanoverschelde@chu-montpellier.fr; 3Department of Anesthesiology and Critical Care, Saint Eloi Hospital, Montpellier University Hospital, 34090 Montpellier, France; s-jaber@chu-montpellier.fr; 4PhyMedExp, University of Montpellier, INSERM (French Institut of Health and Medical Research), CNRS (French National Centre for Scientific Research), 34090 Montpellier, France; m-mourad@chu-montpellier.fr; 5Department of Anesthesiology and Critical Care, Lapeyronie Hospital, Montpellier University Hospital, 34090 Montpellier, France; j-charbit@chu-montpellier.fr; 6Department of Anesthesiology and Critical Care, Gui de Chauliac Hospital, Montpellier University Hospital, 34090 Montpellier, France; k-chalard@chu-montpellier.fr; 7Department of Anesthesiology and Critical Care, Arnaud de Villeneuve Hospital, Montpellier University Hospital, 34090 Montpellier, France

**Keywords:** systemic rheumatic disease, interstitial lung disease, intensive care unit, mortality, outcome

## Abstract

Critically ill patients with systemic rheumatic diseases (SRDs) have a fair prognosis, while those with interstitial lung disease (ILD) have a poorer outcome. However, the prognosis of SRD patients with ILD admitted to the intensive care unit (ICU) remains unclear. We conducted a case–control study to investigate the outcomes of critically ill SRD-ILD patients. Consecutive SRD-ILD patients admitted to five ICUs from January 2007 to December 2017 were compared to SRD patients without ILD. Mortality rates were compared between groups, and prognostic factors were then identified. One hundred and forty critically ill SRD patients were included in the study. Among the 70 patients with SRD–ILD, the SRDs were connective tissue diseases (56%), vasculitis (29%), sarcoidosis (13%), and spondylarthritis (3%). Patients were mainly admitted for acute exacerbation of SRD-ILD (36%) or infection (34%). ICU, in-hospital, and one-year mortality rates in SRD-ILD patients were higher than in SRD patients without ILD (*n* = 70): 40% vs. 16% (*p* < 0.01), 49% vs. 19% (*p* < 0.01), and 66% vs. 40% (*p* < 0.01), respectively. Hypoxemia, high sequential organ failure assessment (SOFA) score, and admission for ILD acute exacerbation were associated with ICU mortality. In conclusion, ILD worsened the outcomes of SRD patients admitted to the ICU. Admissions related to SRD-ILD acute exacerbation and the severity of the acute respiratory failure were associated with ICU mortality.

## 1. Introduction

Systemic rheumatic diseases (SRDs) are a heterogenous group of diseases that share common features such as difficult diagnosis, potential multiorgan involvement, different therapeutic options, including immunosuppressive treatments, and a natural history mainly marked by both acute exacerbations and super-infections [1]. This complexity implies management by a multidisciplinary team, particularly in intensive care unit (ICU) settings [2]. Despite this, several studies have reported that ICU patients with SRDs have fair short- and long-term outcomes [1,3].

On the contrary, patients with interstitial lung disease (ILD) admitted to the ICU are known to have poor outcomes [4,5,6]. Indeed, in a recent review, Huapaya et al. reported a high in-hospital mortality rate at 52% [7]. The outcome seems mainly linked to illness severity, need for mechanical ventilation [5,7], and high-resolution computed tomography (HRCT) patterns, such as fibrosis [8,9] and usual interstitial pneumonia (UIP) [10]. However, the prognosis is also linked to the underlying etiology [11].

Immunosuppressive treatments can modify SRD evolution [12]; therefore, ILD related to SRDs (SRD–ILD) is reputed to be associated with better outcomes than other forms of ILD [11]. However, data on critically ill patients with SRD-ILD are very scarce. The effect of lung involvement on the prognosis of SRD patients admitted to the ICU is unclear and mortality rates and prognosis factors remain questionable in SRD-ILD patients.

The aim of this study was to assess ICU, in-hospital, and one-year mortality rates in SRDs with and without ILD, then to identify factors independently associated with in-ICU mortality in SRD-ILD patients.

## 2. Materials and Methods

### 2.1. Patients

This observational retrospective case–control study was carried out from 1 January 2007 to 31 December 2017 in five ICUs of Montpellier University Hospital. All consecutive SRD-ILD patients admitted to ICUs during the study period were included. Investigators retrieved eligible patients by (i) screening SRD-ILD in the ICU database using the International Classification of Diseases, Tenth Revision (ICD-10) codes of ILD and SRDs (in French: PMSI—programme de medicalisation des systemes d’information) and (ii) by reviewing all ICU discharge summaries. SRD diagnoses were retrospectively and blindly confirmed, then clustered by an intensivist (L.B.), a pneumologist (M.A.), and an internist/immunologist (R.L.), according to the previously published classification criteria [13,14,15,16,17,18,19,20,21]. Furthermore, ILD diagnoses were retrospectively assessed by a pneumologist (M.A.) and a radiologist (J.V.) according to the American Thoracic Society/European Respiratory Society (ATS/ERS) classification [22]. SRD-ILD patients who had typical patterns of diffuse alveolar hemorrhage or cardiac pulmonary edema and those admitted to the ICU after scheduled surgery were excluded. When a patient was admitted twice or more, only the first ICU stay was considered. A control group of critically ill SRD patients without ILD matched on the basis of age, sex, Charlson index, SOFA score, SRD type, and/or level of immunosuppression was set up by extracting patients from the cohort presented by Larcher et al. [3].

### 2.2. Data Collection

Medical records, including discharged summaries, biological results, and medical imagery, were extracted from the institutional medical software (DxCare^®^, DEDALUS FRANCE S.A., Le Plessy-Robinson, France). Reasons for ICU admission were classified into “acute respiratory failure related to a pulmonary disease” and “other reasons” (extra-pulmonary sepsis causing dyspnea, cardiogenic pulmonary edemas, bronchospasms, or other). ICU diagnoses were classified into one of the following categories: infection, SRD-ILD acute exacerbation, and other. According to the definition of idiopathic pulmonary fibrosis acute exacerbation [23], an SRD-ILD acute exacerbation was retained when the following were associated: (i) a previous or concurrent diagnosis of SRD–ILD; (ii) an acute worsening or development of dyspnea typically lasting for less than 30 days; (iii) HRCT with new bilateral ground-glass opacity and/or consolidation superimposed on an underlying pattern consistent with ILD, or new bilateral alveolo-interstitial opacities on the chest X-ray when no HRCT was available; (iv) deterioration not fully explained by cardiac failure or fluid overload. Furthermore, infection diagnosis was retained in patients with clinical and biological signs of infections and after ruling out other diagnoses [24]. All ICU diagnoses were retrospectively and blindly confirmed by an intensivist (L.B.), a pneumologist (M.A.), an internist (R.L.), and a radiologist (J.V.). Whenever a discrepancy between reviewers appeared, diagnosis was discussed between the reviewers until a consensus was achieved.

Demographical data, morbidities, and prior health status were collected. The Charlson index [25], ICU severity scores, simplified acute physiology score II (SAPS II), sequential organ failure assessment (SOFA), SOFA without respiratory system assessment (non-pulmonary SOFA), and Glasgow coma scale were calculated at ICU admission [26,27,28]. During ICU stay, the worst oxygen arterial partial pressure to fraction of inspired oxygen (PaO_2_/FiO_2_) ratio [29], the occurrence of acute kidney injury (AKI) [30], ICU-acquired infections were collected, as the need for vasoactive drugs, mechanical ventilation, renal replacement therapy, and extracorporeal membrane oxygenation (ECMO). Immunosuppressive treatment and corticosteroid therapy before and during ICU stay were recorded.

All thoracic HRCT realized before and after ICU admission were blindly analyzed by a radiologist (J.V.) and a pneumologist (M.A.) and classified by patterns as follows: UIP, nonspecific interstitial pneumonia (NSIP), acute interstitial pneumonia (AIP), or unclassifiable. Additional HRCT patterns of fibrosis or acute lung injury were also identified. Fibrosis pattern was defined by additional lesions such as honeycombing and/or traction bronchiectasis, whereas acute lung injury radiological pattern was defined by acute onset of ground-glass opacities, consolidation, and/or interlobular septal thickening in HRCT. Discrepancies in HRCT between the two reviewers were also resolved by discussion.

### 2.3. Outcomes

Lengths of stay and associated outcomes were collected, including ICU, in-hospital, and one-year after ICU admission mortalities. Vital statistics and date of death, if applicable, were collected using the electronic medical records of each hospital or by phone call to the family doctor, to the patient, or to their next of kin. Mortality rates were assessed and compared to that of a control group of critically ill SRD patients without ILD. Predictive factors of ICU mortality among SRD-ILD patients were then identified.

### 2.4. Statistical Analysis

Data are described as the median and interquartile range (IQR) or number and percentage.

One-year survival was defined as one year starting from ICU admission. The ICU, in-hospital, and one-year mortalities of SRD-ILD patients were compared to a control group of ICU SRD patients without ILD. The one-year survival of these two groups is presented with Kaplan–Meier curves and compared using the log-rank test.

The SRD-ILD population was divided according to vital status in ICU. Categorical variables were compared using Chi-square tests and continuous variables using the nonparametric Wilcoxon test. Independent factors associated with ICU mortality were assessed using a logistic regression model. A conditional stepwise regression with 0.2 as the critical *p*-value for entry into the model was performed to select the most informative variables. Then, given the number of events (28 patients died in ICU), we selected only three variables according to their clinical relevance and statistical significance in the univariate analysis to include in the multivariate analysis [31]. Interactions and correlations between the explanatory variables were carefully checked.

All tests were two-sided and *p*-values less than 0.05 were considered statistically significant. Analyses were done using R software version 3.5.0 (Free Software Foundation, Boston, MA, USA).

### 2.5. Ethical Considerations

The Montpellier University Hospital Institutional Review Board approved this study (no. 2018_IRB-MTP_09-18) and waived the need for written consent. The study was registered at ClinicalTrials.gov (No. NCT04398381) on 21 May 2020.

## 3. Results

### 3.1. General Characteristics of the Study Population

Among 108 patients screened, 70 SRD-ILD patients (38 females) with a median age of 65 (58; 74) years old and a median Charlson index of 4 (3; 6) were included in the study (Figure 1). The demographics and characteristics of these patients are summarized in Table 1.

SRD diagnoses were as follows: 39 patients (56%) had a connective tissue disorder (CTD): 18 (26%) systemic sclerosis, 9 (13%) myositis, 9 (13%) rheumatoid arthritis, 2 (3%) Sjögren’s syndrome, and 1 (1%) systemic lupus erythematosus; 20 patients (29%) had a vasculitis: 12 (17%) granulomatosis with polyangiitis, 4 (6%) microscopic polyangiitis, and 4 (6%) unclassified; 9 patients (13%) had a sarcoidosis, and 2 patients (3%) had an ankylosing spondylitis. Before ICU admission, 38 patients (54%) were treated with corticosteroids and 19 (27%) with immunosuppressive treatments (in association or not). As shown in Table 1, high median SAPS II (43 (32; 59)) and SOFA score (7 (4; 9)) underlined the severity of the patients’ illnesses. Patients were mostly admitted for a pulmonary acute respiratory failure related to a pulmonary disease (39 patients, 56%). Diagnoses retained after ICU stay were mainly SRD-ILD acute exacerbations (25 patients, 36%) or infections (24 patients, 34%). Other diagnoses included cardiogenic pulmonary edema (6 patients), hemorrhagic shock (6 patients), neurologic failure (3 patients), and miscellaneous (6 patients). During ICU stay, 44 patients (63%) required invasive mechanical ventilation, 38 (54%) vasopressive drugs. Twenty-four patients (34%) underwent renal replacement therapy and one required ECMO. More than two-thirds of patients experienced an AKI with 33 (47%) at KDIGO (kidney disease improving global outcomes) stage 3 and 28 patients (40%) had a PaO_2_/FiO_2_ ≤ 100. Most patients were treated by corticosteroids (56 patients, 80%), whereas 18 (26%) received an immunosuppressive treatment, including: cyclophosphamide 500–600 mg/m^2^ IV (12 patients, 1 patient in 2008 and the rest after 2013), rituximab 1 g IV repeated 15 days later (5 patients, 1 patient in 2008 and the rest after 2014), and a combination of tacrolimus and mycophenolate (1 patient). Seven patients (10%) had plasma exchange therapy.

### 3.2. HRCT Patterns

During ICU stay, HRCT was realized at least once in 61 patients (87%). Among them, an NSIP was diagnosed in 17, a UIP in 11, and an AIP in 5. In the remaining 28 patients, HRCT pattern was considered unclassifiable (Table 2). In addition to the underlying ILD pattern, patients may have lesions such as fibrosis and/or a radiological acute lung injury pattern. Forty patients (66%) had a fibrosis pattern and 37 (61%) had a radiological acute lung injury pattern.

### 3.3. Outcomes

Among the SRD-ILD patients, 28 died in the ICU, corresponding to a mortality rate of 40%. The median ICU length-of-stay was 7 (3–20) days. Six additional patients died before hospital discharge, bringing the in-hospital mortality rate to 49% (Table 1 and Figure 1 and Figure 2). Among the 36 patients discharged from hospital, 12 died one year after admission to the ICU, resulting in a mortality rate of 66% (Table 1 and Figure 1 and Figure 2). None of the surviving patients benefited from lung transplantation. We also compared ICU survival between three periods, namely, 2007–2009 (30% (15.4–58.6)), 2010–2013 (41.2% (23.3–72.7)), and 2014–2017 (33.3% (20.6–54.0)), and we did not observe any significant differences between periods (log-rank test, *p* = 0.9).

The demographics and treatment before admission of our 70 SRD-ILD patients were comparable to those of the patients of the control group (Table 3). SRD-ILD patients showed significantly higher ICU, in-hospital, and one-year mortality rates than SRD patients without ILD: 40% vs. 16% (*p* < 0.01), 49% vs. 19% (*p* < 0.01), and 66% vs. 40% (*p* < 0.01), respectively (Table 3 and Figure 2).

### 3.4. Prognostic Factors

In the univariate analysis, SAPS II (*p* = 0.02), SOFA score (*p* = 0.03), need for invasive ventilation (*p* < 0.01), need for vasoactive drug (*p* < 0.01), renal replacement therapy (*p* = 0.04), low PaO_2_/FiO_2_ (*p* < 0.01), and admission for ILD acute exacerbations (*p* < 0.01) were associated with ICU mortality in SRD-ILD patients (Table 1). The occurrence of an acute lung injury pattern in HRCT (*p* = 0.04) was also associated with ICU mortality (Table 2). The type of immunosuppressive therapy was not associated with ICU mortality.

Logistic regression identified a high SOFA score (*p* < 0.01), low PaO_2_/FiO_2_ (*p* < 0.01), and admission for ILD acute exacerbation (*p* < 0.01) as independently associated with ICU mortality in SRD-ILD patients (Table 4).

## 4. Discussion

In this study, we investigated a ten-year cohort of critically ill SRD-ILD patients and compared them to a control group of SRD patients without ILD. Our main finding was that outcomes, particularly long-term outcomes, were poorer in the SRD-ILD group as shown by the ICU, in-hospital, and one-year mortality rates at 40% vs. 16%, 49% vs. 19%, and 66% vs. 40%, respectively. ICU admission for ILD acute exacerbation, higher SOFA score, and lower PaO_2_/FiO_2_ ratio was significantly associated with ICU mortality in SRD-ILD patients.

Previous studies have reported that 10% to 25% of SRD patients are admitted to hospital during the course of their disease, and that one-third of these patients need ICU admission [32]. It is noteworthy that those admitted to the ICU have fair outcomes, similarly to other ICU patients with the same level of illness severity [1,3]. However, our results highlighted the burden of lung involvement in critically ill patients with SRD and showed that ILD dramatically worsens the outcome, bringing them closer to those of other ILD patients.

The outcomes of ILD patients have been reported as poor, especially when requiring invasive mechanical ventilation [6,8,33,34], which has led to considering invasive mechanical ventilation futile in these settings. Indeed, Mallick et al. reviewed the data of patients with idiopathic pulmonary fibrosis who were mechanically ventilated and found an in-hospital mortality of 87% and a three-month mortality of 94% [34]. However, data from the poor outcomes associated with patients with interstitial pulmonary fibrosis requiring invasive ventilation are often assigned confusedly to all patients with ILD [6,9]. Recently, Gannon et al. [11] reported the results of a large cohort of 126 critically ill ILD patients and found that a diagnosis of CTD–ILD was independently associated with a substantially lower risk of short- and long-term death compared to other ILDs. In-hospital and one-year risk of death were 39% and 52% for CTD–ILD patients, whereas those of other ILDs were above 70% and 80%, respectively. Our study showed that almost half of the critically ill SRD-ILD patients did not survive to hospital discharge and, at one year, only one third were alive. Although, the outcomes of our patients seemed slightly worse than those of CTD–ILD patients as reported by Gannon [11], they may be somewhat better than those reported previously in critically ill ILD patients [8,33,34]. Moreover, our cohort was larger, included a different case mix, and was characterized by a higher illness severity (SAPS II at 43 [32; 59] and SOFA at 7 [4,5,6,7,8,9]). It is well known that high SAPS II and/or SOFA score and the need for life-sustaining therapies such as vasoactive drugs and invasive ventilation are associated with poor outcomes in critically ill patients [26,28,35].

Our major concern was to identify predictive factors of mortality to help physicians decide the level and degree of life support in the case of clinical worsening and ICU admission in SRD patients. In our cohort, ICU admission for ILD acute exacerbation was significantly associated with mortality. Severe acute exacerbation seems to be a turning point in the SRD-ILD clinical course, even though the disease potential reversibility may not be discarded. Our observation that the presence of a radiological acute lung injury pattern—potentially related to an acute exacerbation, an infection, or both—was associated with mortality, albeit by univariate analysis, may underline the former assumption. Whether delayed administration of immunosuppressive treatment in such circumstances may jeopardize patient outcomes, as reported in critically ill patients with vasculitis [36], remains questionable. This work was not designed to give any answer to this issue; further studies are mandatory to determine the optimal timing of immunosuppressive treatment and the benefit of innovative therapeutics early after ICU admission [37].

One striking finding of this study was that SRD-ILD patients’ mortality was mainly associated with a higher SOFA score and a lower PaO_2_/FiO_2_ ratio during ICU stay. It is noteworthy that the nonpulmonary SOFA score was not associated with mortality. In addition, the comparison of critically ill SRD patients with and without ILD showed that the former have the poorer outcomes. This suggests that interstitial lung injury and the level of injury severity are among the most determinant in prognosis, such has been already reported in patients admitted for an acute respiratory distress syndrome (ARDS) related to variable causes [29]. In the case of refractory hypoxemia, ECMO may be used as a bridge for transplantation in ILD patients [38]. Nonetheless, lung transplantation remains controversial for these patients mostly with extra-pulmonary organ involvement that may complicate management after surgery [39]. In our cohort, no patient was eligible for lung transplantation, and the only patient who benefited from an ECMO implantation died. Besides hypoxemia, respiratory system compliance is frequently decreased in ILD patients [5], but these data were not recorded from our cohort. One may hypothesize that invasive mechanical ventilation could worsen the course of SRD–ILD, as reported in ARDS patients [35,40,41,42]. Nevertheless, the optimal ventilation strategy and the use of extracorporeal carbon dioxide removal (ECCO_2_R) [43] or ECMO [44] in SRD-ILD patients remain a subject of debate.

The present study has several limitations. First, the study was limited by its retrospective design, introducing bias in data collection and results interpretation. However, ICU admission of SRD-ILD patients is rare and it seems difficult to perform a prospective study in this specific population. Second, during a study period of 10 years, heterogeneity in ICU management and post-ICU care and the use of new rheumatic treatment could substantially modify the prognosis. Nonetheless, this study is one of the largest evaluating short- and long-term outcomes of critically ill SRD–ILD. In addition, our comparison of the mortality rates for three periods did not show any differences. Third, infection or acute exacerbation diagnoses were sometimes not supported by robust evidence and might have been misclassified. Given the severity of the respiratory failure, no patient was able to have a transbronchial biopsy in the ICU. To secure our data collection, all diagnoses were retrospectively and blindly confirmed by four different investigators with a post-hoc analysis of the medical files, including microbiological confirmation when available. Fourth, only patients admitted to the ICU were included in the study, which may have led to selection bias, minimizing the estimation of mortality rates by exclusion of patients denied ICU admission because of do-not-resuscitate management. Lastly, quality of life, readmission, and functional status are important and relevant outcomes but were been collected in this study.

## 5. Conclusions

In conclusion, a diagnosis of ILD significantly worsens the prognosis of SRD patients admitted to the ICU. Critically ill SRD-ILD patients have poor outcomes with an ICU mortality rate at 40% and a one-year after ICU admission rate of 66%. SRD-ILD acute exacerbation and interstitial lung injury severity were the main predictors of ICU mortality. Management of SRD-ILD patients is complex and requires a multidisciplinary approach. Further studies are intended to improve the ICU outcomes of these patients.

## Figures and Tables

**Figure 1 jcm-10-01037-f001:**
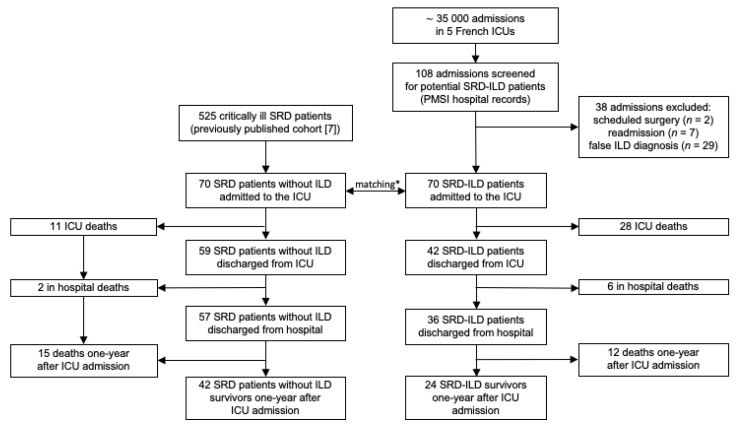
Flow chart of the study population. ICU = intensive care unit, ILD = interstitial lung disease, PMSI = Programme de Medicalisation des Systemes d’Information, SRD = systemic rheumatic diseases. * Control group of critically ill SRD patients without ILD matched on the basis of age, sex, Charlson index, SOFA score, SRD type, and/or level of immunosuppression.

**Figure 2 jcm-10-01037-f002:**
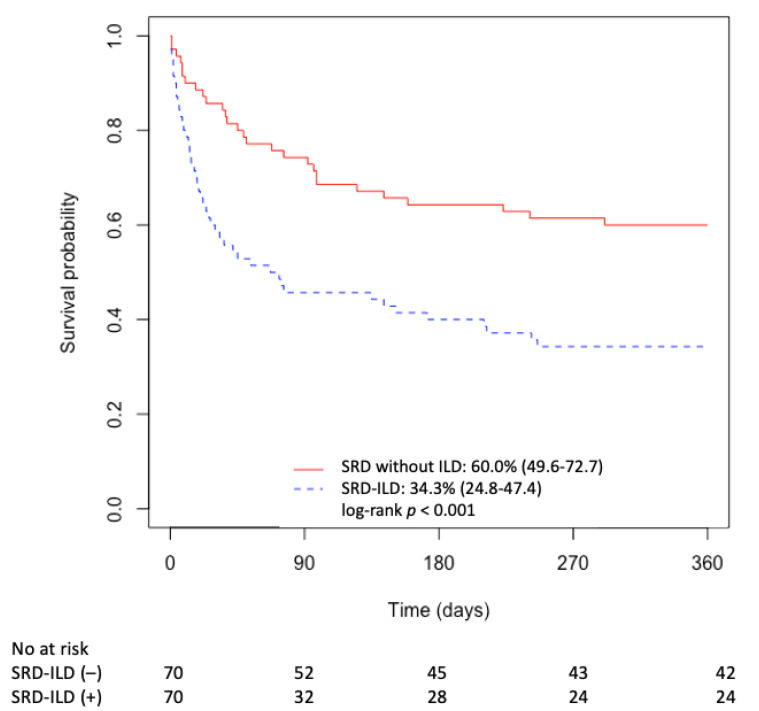
Comparison of Kaplan–Meier curves of one-year survival after ICU admission of SRD patients with (blue dashed line) and without interstitial lung disease (red line). Control group of critically ill SRD patients without ILD matched on the basis of age, sex, Charlson index, SOFA score, SRD type, and/or level of immunosuppression. ICU = intensive care unit, ILD = interstitial lung disease, SRD = systemic rheumatic diseases.

**Table 1 jcm-10-01037-t001:** SRD-ILD patients’ characteristics.

	All Patients *n* = 70	ICU Deceased *n* = 28	ICU Survived *n* = 42	*p*-Value (Univariate Analysis)
Characteristics at admission				
Sex (Male), *n* (%)	32 (45.7)	13 (46.4)	19 (45.2)	1.00
Age (years), median (IQR)	65.0 (58.4–73.7)	63.3 (57.4–70.6)	65.9 (59.2–73.9)	0.49
BMI, median (IQR)	25.1 (21.7–27.0)	25.1 (23.2–26.1)	25.1 (21.1–27.9)	0.90
Charlson index, median (IQR)	4 (3–6)	4 (3–6)	5 (4–6)	0.66
SAPS II, median (IQR)	43 (32–59)	49 (41–66)	40 (30–50)	0.02
SOFA score, median (IQR)	7 (4–9)	8 (5–13)	6 (3–8)	0.03
Non-pulmonary SOFA score, median (IQR)	4 (2–6)	4 (1–10)	4 (2–6)	0.39
Serum creatinine (µmol/L), median (IQR)	146 (102–268)	171 (109–252)	127 (85–322)	0.67
SRD Type				
CTD, *n* (%)	39 (55.7)	19 (67.8)	20 (47.6)	0.05
Vasculitis, *n* (%)	20 (28.6)	4 (14.3)	16 (38.1)	
Sarcoidosis, *n* (%)	9 (12.8)	3 (10.7)	6 (14.3)	
Spondylarthritis, *n* (%)	2 (2.9)	2 (7.1)	0 (0.0)	
Treatment before admission				
Corticosteroids, *n* (%)	38 (54.3)	18 (64.3)	20 (47.6)	0.26
Immunosuppressive, *n* (%)	19 (27.1)	9 (32.1)	10 (23.8)	0.62
Treatment during hospitalization				
Corticosteroids, *n* (%)	56 (80.0)	23 (82.1)	33 (78.6)	0.95
Dose of corticosteroids (mg/kg/day), median (IQR)	1.00 (0.10–1.00)	1.00 (0.11–1.37)	0.50 (0.10–1.00)	0.37
Immunosuppressive, *n* (%)	18 (25.7)	11 (39.3)	7 (16.7)	0.06
Plasma exchange, *n* (%)	7 (10.0)	3 (10.7)	4 (9.5)	1.00
Reason for admission				
Acute respiratory failure related to a pulmonary disease, *n* (%)	39 (55.7)	21 (75.0)	18 (42.9)	0.02
Other, *n* (%)	31 (44.3)	7 (25.0)	24 (57.1)	
Diagnosis				
Infection, *n* (%)	24 (34.3)	9 (32.1)	15 (35.7)	<0.01
Acute exacerbation, *n* (%)	25 (35.7)	16 (57.1)	9 (21.4)	
Other, *n* (%)	21 (30.0)	3 (10.7)	18 (42.9)	
KDIGO stage				
0, *n* (%)	19 (27.1)	6 (21.4)	13 (31.0)	0.26
1, *n* (%)	12 (17.1)	4 (14.3)	8 (19.0)	
2, *n* (%)	6 (8.6)	1 (3.6)	5 (11.9)	
3, *n* (%)	33 (47.1)	17 (60.7)	16 (38.1)	
Worst PaO_2_/FiO_2_				
<100	28 (40.0)	20 (71.4)	8 (19.0)	<0.01
100–199	15 (21.4)	4 (14.3)	11 (26.2)	
≥200	27 (38.6)	4 (14.3)	23 (54.8)	
Organ support				
Vasoactive drugs, *n* (%)	38 (54.3)	21 (75.0)	17 (40.5)	<0.01
Invasive ventilation, *n* (%)	44 (62.8)	26 (92.8)	18 (42.9)	<0.01
Prone positioning, *n* (%)	7 (10.0)	5 (17.9)	2 (4.8)	0.17
Renal replacement therapy, *n* (%)	24 (34.3)	14 (50.0)	10 (23.8)	0.04
Outcomes				
Length of ICU stay (days), median (IQR)	7 (3–20)	11 (4–24)	6 (2–16)	-
ICU-acquired infection, *n* (%)	14 (20.0)	5 (17.8)	9 (21.4)	0.95
Death in ICU, *n* (%)	28 (40.0)	28 (100%)	0 (0%)	-
Death at one year, *n* (%)	46 (65.7)	-	18 (43%)	-

IQR = interquartile range, BMI = body mass index, CTD = connective tissue disease, ICU = intensive care unit, ILD = interstitial lung disease, KDIGO = kidney disease improving global outcomes, PaO_2_/FiO_2_ = arterial partial pressure of oxygen to fraction of inspired oxygen ratio, SAPS II = simplified acute physiology score II, SOFA = sequential organ failure assessment, SRD = systemic rheumatic diseases.

**Table 2 jcm-10-01037-t002:** Radiological features.

	Patients with HRCT *n* = 61	Death in ICU *n* = 22	Alive in ICU *n* = 39	*p*-Value (Univariate Analysis)
ILD radiologic pattern				
Unclassifiable, *n* (%)	25 (41)	6 (27.3)	22 (56.4)	0.05
AIP, *n* (%)	5 (8)	4 (18.2)	1 (2.6)	
NSIP, *n* (%)	17 (28)	8 (36.4)	9 (23.1)	
UIP, *n* (%)	11 (18)	4 (18.2)	7 (17.9)	
Fibrosis, *n* (%)	40 (66)	16 (72.7)	24 (61.5)	0.55
Acute lung injury radiological pattern, *n* (%)	37 (61)	19 (86.4)	18 (46.1)	0.004

AIP = acute interstitial pneumonia, HRCT = high-resolution computed tomography, ICU = intensive care unit, ILD = interstitial lung disease, NSIP = nonspecific interstitial pneumonia, UIP = usual interstitial pneumonia.

**Table 3 jcm-10-01037-t003:** Comparison to the control group (SRD patients without ILD).

	**SRD without ILD** ***n* = 70**	**SRD with ILD** ***n* = 70**	***p*-Value** **(Univariate Analysis)**
Patient characteristics at admission			
Sex (Male), *n* (%)	32 (45,7)	32 (45.7)	1
Age (years), median (IQR)	64.6 (57.48–75.53)	65.0 (58.3–73.6)	0.85
BMI, median (IQR)	25.1 (21.2–28.6)	25.1 (21.7–26.9)	0.91
Charlson score, median (IQR)	5 (3–6)	4 (3–6)	0.99
SOFA Score, median (IQR)	7 (4–9)	7 (4–9)	0.89
SRD Type, *n* (%)			
CTD	48 (68.6)	39 (55.7)	0.04
Vasculitis	10 (14,3)	20 (28.6)	
Sarcoidosis	5 (7.1)	9 (12.8)	
AS	7 (10.0)	2 (2.9)	
Treatment before admission			
Corticosteroids, *n* (%)	46 (65.7)	38 (54.3)	0.23
Immunosuppressive therapy, *n* (%)	25 (35.7)	19 (27.1)	0.36
Prognosis			
Death in ICU, *n* (%)	11 (15.7)	28 (40.0)	<0.01
Death in hospital, *n* (%)	13 (18.6)	34 (48.6)	<0.01
Death at one year, *n* (%)	28 (40.0)	46 (65.7)	<0.01

IQR = interquartile range, BMI = body mass index, CTD = connective tissue disease, ICU = intensive care unit, ILD = interstitial lung disease, SOFA = sequential organ failure assessment, SRD = systemic rheumatic diseases.

**Table 4 jcm-10-01037-t004:** Multivariate analysis to predict ICU mortality in SRD-ILD patients.

Variables	Adjusted OR	CI 95%	*p*-Value
SOFA Score	1.75	(1.16–2.52)	<0.01
Worst PaO_2_/FiO_2_			
>200	-	-	-
101–200	1.73	(0.33–9.05)	0.51
≤100	19.60	(2.41–159.20)	<0.01
Diagnosis in ICU			
Acute exacerbation (reference)	-	-	-
Infection	0.02	(0.0001–0.59)	<0.01
Other	0.01	(0.0001–0.19)	<0.01

Model adjusted to SOFA score, PaO_2_/FiO_2_, SRD type, and diagnosis. CI 95% = 95% confidence interval, ICU = intensive care unit, ILD = interstitial lung disease, OR = odds ratio, PaO_2_/FiO_2_ = arterial partial pressure of oxygen to fraction of inspired oxygen ratio, SOFA = sequential organ failure assessment, SRD = systemic rheumatic diseases.

## Data Availability

The authors consent to share the collected data with others. Data will be provided to qualified investigators free of charge, after careful examination of required documents (summary of the research plan, request form and IRB approval) by the study board of investigators. Data will be available immediately after the main publication and indefinitely.

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
