# Peer review of "Interstitial Lung Disease Worsens Short- and Long-Term Outcomes of Systemic Rheumatic Disease Patients Admitted to the ICU: A Multicenter Study"

_jcm, 2021, doi:10.3390/jcm10051037_

Round 1

Reviewer 1 Report

I think this is a good job to obtain more information about SRD-ILD in relation to ICU admission. 
In my opinion, authors have made a great effort to obtain obvious conclusions. Nevertheles, they offer results which are worthy to consider in a review article. 

There is no mention about lung transplantation for these patients, so it is necessary to complete how many of alive patients underwent lung transplantation. From my point of view it should be desirable to include some information about lung transplantation for SRD-ILD in the discussion. 

I think some information about biopsies samples ( if anyone was performed ) ( transbronchial biopsies in ICU) should be also helpful. 

Author Response

I think this is a good job to obtain more information about SRD-ILD in relation to ICU admission. 
In my opinion, authors have made a great effort to obtain obvious conclusions. Nevertheless they offer results which are worthy to consider in a review article. 

First, we thank the reviewer for his/her interest to this work and helpful observations.

There is no mention about lung transplantation for these patients, so it is necessary to complete how many of alive patients underwent lung transplantation. From my point of view it should be desirable to include some information about lung transplantation for SRD-ILD in the discussion. 

In our cohort, no patient had a lung transplantation. However, we agree with the reviewer, this treatment option needs to be addressed in the results and discussion sections. The text was amended accordingly, please see lines 197-198 p6 and lines 281-284, p9

I think some information about biopsies samples ( if anyone was performed ) ( transbronchial biopsies in ICU) should be also helpful. 

As correctly pointed by the reviewer it will be interesting to analyze results of lung biopsy. However, all patients were admitted in ICU with respiratory failure and consequently none had a transbronchial biopsies in ICU. Those who were mechanically ventilated had a high risk of pneumothorax and those who were not under mechanical ventilation were at risk of worsening. This point has been acknowledged in the discussion section (please see study limits paragraph, lines 301-302, p9)

Reviewer 2 Report

The authors report a 10 year retrospective study ending in 2017 of 70 patients with ILD relating largely to autoimmune disease who were admitted to ICU, and compare outcomes up to one year with 70 control ICU admissions of similar patients without ILD. They conclude that the presence of ILD is an independent risk factor for worse outcomes, although less than for patients with 'pure' ILD. Whilst the study has merits it also has significant limitations, many of which the authors describe in the Discussion. These include:

1 The study was commenced 14 years ago and concluded 4 years ago, making it difficult to be sure that care was standardised throughout this decade. Indeed, as the authors admit, therapeutic options have altered a lot during this time. The authors might consider analysing outcomes by year of admission to see if those treated more recently have better survival.

2 On a related point, several patients were treated with immunosuppressives but numbers are contradictory (was it 18 or 19?) and surprisingly low overall.

3 Most patients on immunosuppression received cyclophosphamide (12) but few details are given of the regime employed. Only 5 patients received rituximab which would now be considered one of the optimal agents. Again, the authors could describe whether year of diagnosis influenced this and whether the choice of agent affected outcome.

4 The data on steroid therapy requires more clarity too. It appears that those patients with ILD who died were more likely to have been on steroids prior to admission and more likely to have received higher does during admission. The authors might like to discuss whether this was associated with increased fatal infection which is well recognised as a complication of long term or high dose steroids

5 There is such heterogeneity in diagnosis, with 39 patients having a CTD (of which systemic sclerosis accounted for nearly half), that it is hard to make sweeping conclusions about the influence of either diagnosis or treatment on outcome. However, the extent of immune disturbance varies greatly from vasculitides with GPA carrying a 100% untreated mortality ,to spondyloarthritis where lung involvement itself is rare and respiratory compromise often a consequence of mechanical chest wall issues. This merits much more discussion to explain and explore these differences.

Overall, if this work is to make a useful contribution to the literature, it needs to be much more specific about the influence of each of the above factors on outcome in order to avoid sweeping generalisations that may be unable to offer useful guidance to clinicians in future.

Author Response

The authors report a 10 year retrospective study ending in 2017 of 70 patients with ILD relating largely to autoimmune disease who were admitted to ICU, and compare outcomes up to one year with 70 control ICU admissions of similar patients without ILD. They conclude that the presence of ILD is an independent risk factor for worse outcomes, although less than for patients with 'pure' ILD. Whilst the study has merits it also has significant limitations, many of which the authors describe in the Discussion. These include:

The authors thank the reviewer for being interested in this work and for his/her thorough comments and remarks.

1 The study was commenced 14 years ago and concluded 4 years ago, making it difficult to be sure that care was standardised throughout this decade. Indeed, as the authors admit, therapeutic options have altered a lot during this time. The authors might consider analysing outcomes by year of admission to see if those treated more recently have better survival.

We totally agree with the reviewer and we have already included this point in the limits section. We compared as suggested by reviewer ICU survival during 3 periods: 2007-2009 (30% [15.4-58.6]), 2010-2013 (41.2% [23.3-72.7]), and 2014-2017 (33.3% [20.6-54]), log-rank test p = 0.9. Survival rates were not significantly different between periods. Please see lines 198-201, p6 and lines 300-301, p9

2 On a related point, several patients were treated with immunosuppressives but numbers are contradictory (was it 18 or 19?) and surprisingly low overall.

Indeed, numbers may appear contradictory. In fact, patients were treated with immunosuppressives before ICU admission and 18 had an immunosuppressive therapy initiated during ICU stay. Please see Table 1. A majority of patients were treated by corticoids before ICU admission and all of patients with SRD flair-up received corticoids or an immunosuppressive therapy during ICU stay.

3 Most patients on immunosuppression received cyclophosphamide (12) but few details are given of the regime employed. Only 5 patients received rituximab which would now be considered one of the optimal agents. Again, the authors could describe whether year of diagnosis influenced this and whether the choice of agent affected outcome.

We thank the reviewer for his/her accurate comment. The regimen employed were as follow: Cyclophosphamide 500-600 mg/m2 IV and Rituximab 1 g IV, repeated two weeks later.

4 The data on steroid therapy requires more clarity too. It appears that those patients with ILD who died were more likely to have been on steroids prior to admission and more likely to have received higher does during admission. The authors might like to discuss whether this was associated with increased fatal infection which is well recognised as a complication of long term or high dose steroids

In our study, corticosteroid treatment before or after ICU admission was not associated with mortality. Indeed, patients who died in ICU received higher doses of steroids (1mg/kg vs. 0.5 mg/kg) but this was not significantly associated with mortality. However, by multivariate analysis ICU admission for an ILD acute exacerbation was an independently associated with mortality. Please see Tables 1 and 4.

5 There is such heterogeneity in diagnosis, with 39 patients having a CTD (of which systemic sclerosis accounted for nearly half), that it is hard to make sweeping conclusions about the influence of either diagnosis or treatment on outcome. However, the extent of immune disturbance varies greatly from vasculitides with GPA carrying a 100% untreated mortality ,to spondyloarthritis where lung involvement itself is rare and respiratory compromise often a consequence of mechanical chest wall issues. This merits much more discussion to explain and explore these differences.

We agree with the reviewer, as in numerous studies on ILDs in the ICU, the case mix is heterogeneous. However, all ILD diagnoses were retrospectively assessed by a pneumologist (M.A.) and a radiologist (J.V.) according to the ATS/ERS classification (line 71-72 p2). Patients with spondyloarthritis included in our cohort have ILD per se, and those with restrictive lung disease related spondyloarthritis have been excluded. For patients with ANCA vasculitis, only those with ILD per se have been also included, and those with diffuse alveolar hemorrhage have been excluded. Finally, no association between short or long-term outcomes and SRD type has been found by uni- and multivariate analysis, suggesting regardless SRD type, the prognosis is burdened by lung involvement.

Overall, if this work is to make a useful contribution to the literature, it needs to be much more specific about the influence of each of the above factors on outcome in order to avoid sweeping generalisations that may be unable to offer useful guidance to clinicians in future.

We hope that we have answered your questions thoroughly and have complied with your recommendations in order to improve the quality and the meaning of our work to a significant extent.

Round 2

Reviewer 1 Report

The authors have made an important effort to answer all the questions and I believe it is ready to be published in JCM

Author Response

Thank you for your comments and your help. 

Reviewer 2 Report

Thank you for appreciating my constructive criticism and for answering the points I made. Within the confines associated with this type of study, you have addressed the main issues that I had concerns about although I would still like to know more about the years in which Rituximab (n=5) and cyclophosphamide were administered, and whether this influenced survival. I have identified no new concerns.

Author Response

Thank you again for your help. Among the 5 patients who received rituximab in ICU, one was treated in 2008, all the others the five last years of the study period (2014-2017). We don't find any association between rituximab and mortality (only 1 survival/5). Twelve patients received cyclophosphamide during ICU stay and all but one patient were treated after 2013. Four patients survived. We amended the text accordingly, please see lines 179-180, p5 and lines 221-222, p7.